# Healthcare providers' perception towards utilization of health information applications and its associated factors in healthcare delivery in health facilities in Cape Coast Metropolis, Ghana

Richard Okyere Boadu[1]*, Godwin Adzakpah[1], Nathan Kumasenu Mensah[1], Kwame Adu Okyere Boadu[2], Jonathan Kissi[1], Christiana Dziyaba[3], Rosemary Bermaa Abrefa[1]

1 Department of Health Information Management School of Allied Health Sciences, College of Health and Allied Health Sciences, University of Cape Coast, Cape Coast, Ghana, 2 School of Medicine and Dentistry, College of Health Sciences, Kwame Nkrumah University of Science and Technology, Kumasi, Ghana, 3 Health Information Management Unit, Adisadel Health Centre, Ghana Health Service, Accra, Ghana

* richard.boadu@ucc.edu.gh

## Abstract

### Background

Information and communication technology (ICT) has significantly advanced global healthcare, with electronic health (e-Health) applications improving health records and delivery. These innovations, including electronic health records, strengthen healthcare systems. The study investigates healthcare professionals' perceptions of health information applications and their associated factors in the Cape Coast Metropolis of Ghana's health facilities.

### Methods

We used a descriptive cross-sectional study design to collect data from 632 healthcare professionals (HCPs), in the three purposively selected health facilities in the Cape Coast municipality of Ghana, in July 2022. Shapiro-Wilk test was used to check the normality of dependent variables. Descriptive statistics were used to report means with corresponding standard deviations for continuous variables. Proportions were also reported for categorical variables. Bivariate regression analysis was conducted to determine the factors influencing the Benefits of Information Technology (BoIT); Barriers to Information Technology Use (BITU); and Motives of Information Technology Use (MoITU) in healthcare delivery. Stata SE version 15 was used for the analysis. A *p*-value of less than 0.05 served as the basis for considering a statistically significant accepting hypothesis.

### Results

Healthcare professionals (HCPs) generally perceived moderate benefits (Mean score (M) = 5.67) from information technology (IT) in healthcare. However, they slightly agreed that barriers like insufficient computers (M = 5.11), frequent system downtime (M = 5.09), low

**Data Availability Statement:** All relevant data are within the paper and its Supporting information files.

**Funding:** This study was self-funded as part of academic work.

**Competing interests:** The authors have declared that no competing interests exist.

system performance (M = 5.04), and inadequate staff training (M = 4.88) hindered IT utilization. Respondents slightly agreed that training (M = 5.56), technical support (M = 5.46), and changes in work procedures (M = 5.10) motivated their IT use. Bivariate regression analysis revealed significant influences of education, working experience, healthcare profession, and IT training on attitudes towards IT utilization in healthcare delivery (BoIT, BITU, and MoITU). Additionally, the age of healthcare providers, education, and working experience significantly influenced BITU. Ultimately, age, education, working experience, healthcare profession, and IT training significantly influenced MoITU in healthcare delivery.

## Conclusions

Healthcare professionals acknowledge moderate benefits of IT in healthcare but encounter barriers like inadequate resources and training. Motives for IT use include staff training and support. Bivariate regression analysis shows education, working experience, profession, and IT training significantly influence attitudes towards IT adoption. Targeted interventions and policies can enhance IT utilization in the Cape Coast Metropolis, Ghana.

## Introduction

Information and Communication Technology (ICT) has long been acknowledged for its significant role, influence, and impact on all aspects of our society. ICT tools are increasingly being developed, advocated, and used in the health sector to improve the quality of work in administration, patient records, health services, and research. Health Professionals and medical professionals have kept extensive notes on diseases encountered and therapies applied in all areas of medicine on paper for thousands of years. This technique of preserving health records on paper is vulnerable to security threats, offers limited storage space, and thus does not ease the transition of decision-making analysis [1].

Most facets of healthcare are accelerated and advanced by ICT in the global health sector. These include the usage of electronic medical records, virtual office visits, arranging and paying for appointments online, and receiving medicine prescriptions electronically [2]. Rouleau, Gagnon, and Côté found that information technology in healthcare delivery is largely beneficial for the continuing professional growth of Healthcare Providers (HCPs) [3]. Using Electronic Health Records, HCPs, particularly Health Information Managers, are better equipped to connect and relate with patients. The use of Information Technology in Electronic health record (EHR) systems allows for the capturing, storing and sharing of a patient's personal as well as medical information [4]. The importance of ICT in improving overall health record management cannot be overstated. The ability of ITs to capture, store, retrieve, analyze, and send enormous amounts of health records across several locations demonstrates this [5]. The use of ICT in healthcare delivery, also known as E-health, has been lauded for not only improving the reliability and effectiveness of health records and health delivery but also for strengthening healthcare delivery systems through various tailor-made innovative applications and programs like electronic health records [6]. The implementation of ICT applications in healthcare delivery, especially in developing countries, faces significant challenges. These challenges include high Initial Costs, barriers to Use, Patient Privacy and Security, some healthcare providers view information technology as a management tool rather than a clinical care tool, lack of availability of information technology systems, and inadequate training for healthcare professionals [7].

The Ministry of Health in Ghana adopted the Government of Ghana's ICT policy document in 2005 and the National e-health Strategy in 2010 [8]. The objectives of this strategy were to; improve ICT infrastructure in the health sector; improve access to and management of health information; improve access to quality health using telemedicine; and improve ICT knowledge, capability and utilization among health workers. Ghana is growing significantly with the development of e-Health in its national healthcare delivery [9]. Many ICT initiatives have sprung up including the District Health Information Management System (DHIMS II) and MOTECH and National Health Insurance Authority's (NHIA) initiatives. The District Health Information Management System (DHIMS II) is a comprehensive health information system for capturing, reporting and analysing health data for the healthcare ecosystem in Ghana. The system has been operational since 2012 and is available in all 216 health districts in Ghana. The MOTECH Platform can be used for setting appointments, tracking any scheduled activity, and managing workers deployed in the field. Its initial implementations have been for mHealth projects that improve health by sending messages to patients and caregivers based on an evaluation of the recommended schedule of care compared to the patient's health-related actions. The National Health Insurance Authority's (NHIA) transition towards a paperless system is on track in line with the national digitization drive. Mobile renewal platform, Claim-it, Electronic cash payments and online Credentialing are the latest innovations gaining ground. A greater number of National Health Insurance Scheme (NHIS) members are switching to the Mobile Renewal Service platform to renew their membership. All NHIS members are required to dial a dedicated short code (*929#) on their mobile phones and pay from their mobile money wallets to stay active. Despite the remarkable growth in the incorporation of ICT in healthcare delivery, some ICT projects do not usually survive beyond the pilot phase [10].

Ghana's Health Record Management sector has not been able to tap into the full potential of modern information technology to improve healthcare delivery due to the lack of re-training of Healthcare Professionals [11]. Access to patients' longitudinal records are often times difficult and cumbersome. The lack of proper access has cost the healthcare industry a huge fortune every year due to duplication and waste [11]. In spite of these challenges, Waema [12] and Knuth et al. [13], advocate for improving the ICT infrastructure in the health sector to facilitate the adoption of e-health solutions. The Ghana Health Service implemented the Lightwave Hospital Information Management System (LHIMS) to digitize patient records and improve the continuity of care in the Cape Coast metropolis. Challenges included storage limitations, missing records, delays in retrieval, network issues, inadequate training, and high folder procurement costs. LHIMS provides accurate, up-to-date patient information, reduces costs, enables quick access to records for coordinated care, and improves diagnoses and safety. Despite its positive impact, interruptions in power supply and unreliable internet connectivity affect workflow, and the system's success depends on user acceptance and utilization. This study assesses HCPs' perceptions towards the utilization of health information applications in healthcare delivery in the Cape Coast Metropolis of Ghana.

Hypotheses

H1: HCP working experience will have a significant influence on BoIT.

H2: HCP profession will have a significant influence on BoIT.

H3: HCP education will have a significant influence on BoIT.

H4: HCP working experience will have a significant influence on BITU.

H5: HCP profession will have a significant influence on BITU.

H6: HCP education will have a significant influence on BITU.

H7: HCP working experience will have a significant influence on MoITU.

H8: HCP profession will have a significant influence on MoITU.

H9: HCP education will have a significant influence on MoITU.

H10: HCP IT training will have a significant influence on BoIT.

H11: HCP IT training will have a significant influence on BITU.

H12: HCP IT training will have a significant influence on MoITU

## Research methods

### Study design

The research used a descriptive cross-sectional design to collect data from HCPs, in the three purposively selected health facilities in the Cape Coast municipality of Ghana, in July 2020. The study involves 632 participants (irrespective of their age, gender, location, affiliation, level of fitness, intellectual ability, etc.) who collect or manage patients' records, in the selected health facilities.

### Study site

The Cape Coast Teaching Hospital is one of the agencies under the Ministry of Health. With a current bed capacity of 400, the hospital is mandated to provide tertiary clinical services, serve as a training for graduate and post graduate medical programs and to undertake research into emerging health problems. It also serves as the referral facility for the health facilities in the Central, Western and Western North regions of Ghana. Established in August 1998 as the Central Regional Hospital and later upgraded to a Teaching Hospital status in March 2014, following the establishment of the School of Medical Science at the University of Cape Coast, Ghana. Cape Coast Teaching Hospital is also accredited for postgraduate training by the Ghana College of Physicians and Surgeons. The hospital is the main training centre for students of the School of Medical Sciences of the University of Cape Coast. It also collaborates with other schools and colleges including the School of Nursing and Midwifery as well as the School of Health and Allied Sciences. These schools train students at both undergraduate and postgraduate levels. The hospital is geographically located in the northern part of Cape Coast and bounded on the North by Abura Township, on the South by Pedu Estate and 4th Ridge, Nkanfoa on the East and Abura/Pedu Estate on the West. Cape Coast Metropolitan Hospital is one of the hospitals with multispecialty departments in the Central Region of Ghana. It was established in 1939 as a district hospital and later upgraded to a regional hospital for the Central Region. In 1998, when the new regional hospital was built, it reverted to the district hospital status. It provides a wide range of health care services to the community in the Central Region. This hospital operates under the auspices of the Ministry of Health and the Ghana Health Service. It has a bed capacity of 115 and 327 clinical and non-clinical staff. It offers outpatient and inpatient services, as well as emergency services. Ewim Polyclinic operates under the Ministry of Health, Ghana. The facility provides outpatient services to Ewim and its surrounding communities. The health post was upgraded to Ewim Clinic in 1986 and later to Ewim Urban Health Center.

### Study population

The study population involves HCPs in the three (3) selected government health facilities in Cape Coast municipality. The facilities include the Cape Coast Teaching Hospital (CCTH), Ewim Polyclinic (EP), and Cape Coast Metro Hospital (CCMH).

## Sample size determination

A total sample size of 717 was selected from an estimated 2112 HCPs from the three (3) health facilities using StatCalc (for population survey or descriptive study) function in EpiInfo 7 software [Confidence level = 99.9%, expected frequency = 50%, an acceptable margin of error = 5%, design effect = 1.0, at a desired power (0.84 for 80% power), cluster = 3]. The estimated sample sizes for the selected facilities are distributed as follows: CCTH = 520, EP = 107, and CCMH = 90.

## Sampling procedures

A purposive sampling was used to select the three health facilities in the Cape Coast municipality. These facilities were selected and included in the study because they have deployed patients' electronic health record systems to render healthcare delivery to their clients. To give equal opportunity to all eligible staff to participate in the study in the chosen facilities, a simple random sampling was used to select participants. At each department/unit a list of staff with serial ID was generated in consecutive order (e.g. 001, 002, . . ., N), where N is the total number of HCPs. A mobile app random generator was used to select participants based on their serial ID until the study population was covered. Staff who were available and selected; and voluntarily consented to participate were included in the study.

## Data collection tools and procedures

A structured questionnaire was adapted from a previous study [14] and was used to solicit information from respondents. The first part of the questionnaire included questions regarding sociodemographic data such as gender, age, education, profession, number of years of experience, and IT training. The second part consists of questions regarding the benefit of IT, barriers to IT use, and motives for IT use are measured. Our assessment indicators consisted of 24 items that measured the "benefit of information technology (BoIT)" (12 items), "barriers to information technology use (BITU)" (8 items), and "motives of information technology use (MoITU)" (4 items). The response scale for all items was a seven-point, positively packed Likert scale [15,16] coded as, 7: Strongly agree; 6: Moderately agree; 5: Slightly agree; 4: Neutral; 3: Slightly disagree; 2: Moderately disagree; 1: Strongly disagree. The participants were interviewed in English which is the official language in the hospitals. About 632 interviews were conducted in private consultation rooms, staff lounges or break rooms in the various Departments/Units within the hospital to ensure privacy, comfort, and convenience for the participants.

## Analysis

Descriptive statistics such as means and standard deviations were calculated for continuous variables while frequencies and percentages were determined for categorical variables. All negative responses were reversed before undertaking analysis for study constructs. The normality of data was assessed using the Shapiro-Wilk test [17]. Statistical reliability test of the variables in the dataset was assessed using Cronbach's alpha reliability coefficient. This method was applied to assess the internal consistency of the survey items [16,18]. The overall Cronbach's alpha was 0.90 indicating high reliability. Individual Cronbach's alpha for various dimensions in the study were also determined. Content Validity Index (CVI) and Content Validity Ratio (CVR) for the tool were also determined. Details of the reliability and validity test are shown in Table 1. Bivariate regression analysis with robust standard error was conducted to determine the factors influencing Benefits of Information Technology (BoIT); Barriers to Information

**Table 1. Descriptive statistics, scale and item reliability test.**

| Dimensions | Number of Items | Health Professionals (n = 632) | | | Cronbach's Alpha Coefficient | Content Validity Index | Content Validity Ratio |
|---|---|---|---|---|---|---|---|
| | | Mean | SD | Mean Values under 95% CI | | | |
| Overall Tool | 24 | | | | 0.90 | 95.8 | 100.0 |
| Benefits of IT | 12 | 5.67 | 0.98 | [5.60–5.75] | 0.90 | | |
| Barriers to IT use | 8 | 4.70 | 0.98 | [4.62–4.77] | 0.79 | | |
| Motives to IT use | 4 | 5.39 | 1.12 | [5.30–5.47] | 0.77 | | |

CI—Confidence Interval; SD–Standard deviation.

Technology Use (BITU); and Motives of Information Technology Use (MoITU) in healthcare delivery. Stata SE version 15 was used for the analysis. A *p*-value of less than 0.05 served as the basis for considering a statistically significant accepting hypothesis.

## Results

### Background characteristics of respondents

Table 2 shows the background characteristics of the study participants. A total of 632 HCPs participated in the study with an 88% response rate which has positive statistical implications, including increased representativeness, improved statistical power and greater confidence in the results. The majority of HCPs 351(55.5%) were females, while 281 (44.5%) were males. Out of the 632 HCPs, 220 (34.8%) were in their 20–29 years, 261 (41.3%) were in their 30–39 years, 120 (19.0%) were in their 40–49 years, and 31 (4.9%) were in the ages of 50 years and above. The range of ages was 20 to 65, with a mean age of 33.8 years and SD of 8.1. The majority 209 (33.1%) of the participants had completed postsecondary education, while 154 participants (24.4%) had obtained bachelor's degrees. A total of 143 (22.6%) HCPs had a diploma, 113 (17.9%) had a master's degree, and 13 (2.1%) had completed a Junior High or Senior High School. A total of 275 (43.5%) HCPs were nurses, followed by 67 (10.6%) midwives, 58 (9.2%) doctors, 44 (6.9%) health information officers/biostatisticians, 37 (5.9%) nutrition officers, 35 (5.5%) disease control officers, 28 (4.4%) dispensary technicians, and 25(4.0%) pharmacists. The remaining participants included physician assistants, lab technicians, and other facility workers. With regards to HCPs working experience, about, 219 (34.7%) respondents had worked for less than 5 years, 194 (30.7%) for 5–9 years, 151 (23.9%) for 10–14 years, 56 (8.9%) for 15–19 years, and 12 (1.9%) for 20 years or more. The majority of HCPs, 579 (91.6%) had some IT training prior to the study.

### Healthcare providers' perceived benefits of information technology in healthcare delivery

The findings reveal that HCPs moderately agreed that, there is easier access to patient records (M = 5.88), there is an easy way to find investigation results (5.86) and helps in preparing hospital reports (5.86) (S1). Again, it was reported that on average most participants slightly agreed that, IT helps in managing patients (M = 5.60), provides speed to accomplish work (M = 5.76), saved paperwork (M = 5.81), facilitates coordination among departments (M = 5.53), improves decisions making process (M = 5.61), and ensures patients'privacy (M = 5.82). However, HCPs slightly agree that IT reduces medical errors (M = 5.36), or improves the quality of patients' care (M = 5.49) and decreases workload (M = 5.46).

**Table 2. Demographic characteristics of study participants.**

| Characteristics | Number of Healthcare Professionals | Percent |
|---|---|---|
| Age* | 33.84 ± 8.16 | |
| Age group | | |
| 20–29 years | 220 | 34.81 |
| 30–39 years | 261 | 41.30 |
| 40–49 years | 120 | 18.99 |
| 50 plus years | 31 | 4.91 |
| Sex | | |
| Female | 351 | 55.54 |
| Male | 281 | 44.46 |
| Education | | |
| 10 years | 13 | 2.06 |
| Intermediate (11–12 years) | 209 | 33.07 |
| Bachelor (13-14years) | 154 | 24.37 |
| Master | 113 | 17.88 |
| Other professional diploma/degree | 143 | 22.63 |
| Working Experience | | |
| Less than 5 years | 219 | 34.65 |
| 5–9 years | 194 | 30.70 |
| 10–14 years | 151 | 23.89 |
| 15–19 years | 56 | 8.86 |
| 20 plus years | 12 | 1.90 |
| Profession | | |
| Doctors/Physicians | 81 | 12.82 |
| Nurses/Midwives | 342 | 54.11 |
| Pharmacists/Disp Tech | 53 | 8.39 |
| Allied Health Professionals | 156 | 24.68 |
| Had Training in IT | | |
| No | 53 | 8.39 |
| Yes | 579 | 91.61 |

Data are presented as frequency and percentage.

*Mean ± Standard deviation.

Generally, HCPs moderately (M = 5.67) perceived that on average they benefit from information technology (S1).

## Factors influencing Benefits of Information Technology (BoIT) in healthcare delivery

The relationship between IT training, other independent variables and BoIT in healthcare delivery are presented in Table 3. The results suggest that education, working experience, healthcare profession and training in IT had a statistically significant influence on BoIT in healthcare delivery. Healthcare professionals with a bachelor (coefficient = 0.44; $p$-value<0.001), masters (coefficient = 0.47; $p$-value<0.001) and other professional diploma/degrees (coefficient = 0.48; $p$-value<0.001) had an increasing effect on BoIT in healthcare delivery compared with those with intermediate certificates. Considering the working experience of healthcare professionals, those who have 5–9 years (coefficient = 0.23; $p$-value = 0.016)

**Table 3. Bivariate regression analysis of factors influencing Benefits of Information Technology (BoIT) in healthcare delivery.**

| Characteristics | Coefficient | Robust SE | [95% CI] | P-value |
|---|---|---|---|---|
| Age group | | | | |
| 20–29 years | Ref | | | |
| 30–39 years | 0.06 | 0.09 | [-0.11–0.22] | 0.524 |
| 40–49 years | 0.15 | 0.11 | [-0.06–0.36] | 0.155 |
| 50 plus years | -0.31 | 0.25 | [-0.81–0.19] | 0.222 |
| Sex | | | | |
| Female | Ref | | | |
| Male | 0.06 | 0.08 | [-0.09–0.21] | 0.433 |
| Education | | | | |
| Intermediate (11–12 years) | Ref | | | |
| 10 years | -0.11 | 0.45 | [-0.99–0.76] | 0.802 |
| Bachelor (13-14years) | 0.44 | 0.10 | [0.25–0.63] | **<0.001**** |
| Master | 0.47 | 0.11 | [0.26–0.69] | **<0.001**** |
| Other professional diploma/degree | 0.48 | 0.11 | [0.27–0.70] | **<0.001**** |
| Working Experience | | | | |
| Less than 5 years | Ref | | | |
| 5–9 years | 0.23 | 0.09 | [0.04–0.41] | **0.016*** |
| 10–14 years | 0.30 | 0.10 | [0.10–0.50] | **0.003*** |
| 15–19 years | -0.22 | 0.16 | [-0.54–0.09] | 0.163 |
| 20 plus years | -0.60 | 0.35 | [-1.29–0.09] | 0.089 |
| Profession | | | | |
| Nurses/Midwives | Ref | | | |
| Doctors/Physicians | 0.34 | 0.09 | [0.16–0.52] | **<0.001**** |
| Pharmacists/Disp Tech | 0.12 | 0.17 | [-0.22–0.45] | 0.495 |
| Allied Health Professionals | 0.27 | 0.09 | [0.09–0.44] | **0.003*** |
| Had Training in IT | | | | |
| No | Ref | | | |
| Yes | 0.69 | 0.19 | [0.31–1.06] | **<0.001**** |

*p-value<0.05 and

**p-value<0.001 are statistically significant.

SE—Standard Error; CI—Confidence Interval.

and 10–14 years (coefficient = 0.30; p-value = 0.003) also had an increasing effect on BoIT in healthcare delivery compared with those with less than 5 years. Similarly, healthcare providers who are doctors/physicians (coefficient = 0.34; p-value<0.001) and allied health professionals (i.e. health information officers, biomedical scientists, nutritionists, etc.) (coefficient = 0.27; p-value = 0.003) had an increasing influence on BoIT in healthcare delivery compared with those with professionals who are nurses/midwives. Regarding training in IT, the healthcare providers with IT training (coefficient = 0.69; p-value<0.001) had a positive association with BoIT in healthcare delivery compared with those without IT training.

H1: HCP working experience had a significant influence on BoIT.

H2: HCP profession had a significant influence on BoIT.

H3: HCP education had a significant influence on BoIT.

H10: HCP IT training had a significant influence on BoIT.

## Healthcare providers' perceived barriers to information technology use

The results from S1 reveal that, on average, HCPs were not sure (M = 4.69) as to whether they had encountered barriers to IT usage. That is, participants were not sure whether lack of technical support (M = 4.80), incapability of the system (M = 4.73), and lack of management support (M = 4.79) are barriers to the use of information technology. More so, participants moderately disagreed that time consumption (M = 3.10) was a barrier to the use of information technology. However, participants slightly agreed that the insufficient number of computers (M = 5.11), the system being down frequently (M = 5.09), low system performance (5.04), and lack of training for the hospital staff (M = 4.88) are the barrier barriers they face for the use of information technology (S1).

## Factors influencing Barriers to Information Technology Use (BITU) in healthcare delivery

The relationship between IT training, other independent variables and BITU in healthcare delivery are presented in Table 4. The results suggest that, age of healthcare provider, education, and working experience had a statistically significant influence on BITU in healthcare delivery. Compared with healthcare providers with aged 20–29 years, those with 50 years and above (coefficient = -0.84; $p$-value<0.001) have a decreasing influence on BITU in healthcare delivery. This suggest that, the older you become, the less impact you perceived the challenges to BITU in healthcare delivery especially when you have built antidote to that challenges that confront you in the workplace. Regarding education, healthcare providers who have master's degree (coefficient = 0.33; $p$-value = 0.004) had an increasing influence on BITU in healthcare delivery compared with those with intermediate certificate. Similarly, healthcare providers with working experience of 5–9 years (coefficient = 0.21; $p$-value = 0.030) had a positive association with BITU in healthcare delivery compared to those with less than 5 years.

H4: HCP working experience had a significant influence on BITU.

H5: HCP profession had no significant influence on BITU.

H6: HCP education had a significant influence on BITU.

H11: HCP IT training had no significant influence on BITU.

## Healthcare providers' perceived motive of information technology use in healthcare delivery

The results indicate that on average the participants slightly disagreed that, the provision of new/durable applications (M = 3.10), and slightly agreed to provide training to staff (M = 5.56), and provision of technical support (M = 5.46) is the motive of information technology use in healthcare delivery (S1). On average, health professionals slightly agreed that changing hospitals' work procedures (M = 5.09) is a motive for information technology use in healthcare delivery. Generally, on average, health professionals had a good motive for information used in healthcare delivery.

**Table 4. Bivariate analysis of factors influencing Barriers to Information Technology Use (BITU) in healthcare delivery.**

| Characteristics | Coefficient | Robust SE | [95% CI] | P-value |
|---|---|---|---|---|
| Age group | | | | |
| 20–29 years | Ref | | | |
| 30–39 years | 0.04 | 0.09 | [-0.14–0.21] | 0.681 |
| 40–49 years | 0.01 | 0.11 | [-0.21–0.21] | 0.990 |
| 50 plus years | -0.84 | 0.17 | [-1.17 - -0.51] | <**0.001**** |
| Sex | | | | |
| Female | Ref | | | |
| Male | -0.02 | 0.08 | [-0.17–0.14] | 0.835 |
| Education | | | | |
| Intermediate (11–12 years) | Ref | | | |
| 10 years | -0.41 | 0.25 | [-0.90–0.08] | 0.103 |
| Bachelor (13-14years) | 0.11 | 0.11 | [-0.10–0.32] | 0.290 |
| Master | 0.33 | 0.11 | [0.10–0.55] | **0.004*** |
| Other professional diploma/degree | 0.08 | 0.11 | [-0.14–0.30] | 0.466 |
| Working Experience | | | | |
| Less than 5 years | Ref | | | |
| 5–9 years | 0.21 | 0.10 | [0.02–0.40] | **0.030*** |
| 10–14 years | 0.10 | 0.11 | [-0.11–0.31] | 0.364 |
| 15–19 years | 0.07 | 0.14 | [-0.21–0.35] | 0.636 |
| 20 plus years | -0.24 | 0.31 | [-0.86–0.38] | 0.442 |
| Profession | | | | |
| Nurses/Midwives | Ref | | | |
| Doctors/Physicians | 0.14 | 0.10 | [-0.06–0.35] | 0.176 |
| Pharmacists/Disp Tech | -0.09 | 0.17 | [-0.42–0.24] | 0.584 |
| Allied Health Professionals | 0.13 | 0.09 | [-0.04–0.30] | 0.129 |
| Had Training in IT | | | | |
| No | Ref | | | |
| Yes | 0.26 | 0.20 | [-0.13–0.65] | 0.188 |

*p-value<0.05 and

**p-value<0.001 are statistically significant.

SE—Standard Error; CI—Confidence Interval.

## Factors influencing Motives of Information Technology Use (MoITU) in healthcare delivery

The relationship between IT training, other independent variables and MoITU in healthcare delivery are presented in Table 5. The analysis revealed that age of healthcare provider, education, working experience, healthcare profession and training in IT had a statistically significant influence on MoITU in healthcare delivery. Compared with healthcare providers with aged 20–29 years, those with 50 years and above (coefficient = -0.66; p-value = 0.003) had a decreasing influence on MoITU in healthcare delivery. Healthcare professionals with a bachelor (coefficient = 0.49; p-value<0.001), masters (coefficient = 0.63; p-value<0.001) and other professional diploma/degrees (coefficient = 0.26; p-value<0.035) had an increasing effect on MoITU in healthcare delivery compared with those with intermediate certificates. Regarding working experience of healthcare professionals, those who have 5–9 years (coefficient = 0.35; p-value<0.001) had an increasing effect on MoITU in healthcare delivery compared with

**Table 5. Bivariate analysis of factors influencing Motives of Information Technology Use (MoITU) in healthcare delivery.**

| Characteristics | Coefficient | Robust SE | [95% CI] | P-value |
|---|---|---|---|---|
| Age group | | | | |
| 20–29 years | Ref | | | |
| 30–39 years | 0.10 | 0.10 | [-0.10–0.31] | 0.320 |
| 40–49 years | 0.09 | 0.12 | [-0.14–0.32] | 0.448 |
| 50 plus years | -0.66 | 0.22 | [-1.09 - -0.23] | **0.003*** |
| Sex | | | | |
| Female | Ref | | | |
| Male | -0.02 | 0.09 | [-0.20–0.16] | 0.828 |
| Education | | | | |
| Intermediate (11–12 years) | Ref | | | |
| 10 years | -0.51 | 0.38 | [-1.25–0.23] | 0.178 |
| Bachelor (13-14years) | 0.49 | 0.12 | [0.27–0.72] | **<0.001**** |
| Master | 0.63 | 0.13 | [0.38–0.88] | **<0.001**** |
| Other professional diploma/degree | 0.26 | 0.12 | [0.02–0.50] | **0.035*** |
| Working Experience | | | | |
| Less than 5 years | Ref | | | |
| 5–9 years | 0.35 | 0.11 | [0.13–0.56] | **<0.001**** |
| 10–14 years | 0.22 | 0.12 | [-0.02–0.46] | 0.068 |
| 15–19 years | 0.07 | 0.17 | [-0.26–0.40] | 0.672 |
| 20 plus years | -0.88 | 0.35 | [-1.57 - -0.20] | **0.012*** |
| Profession | | | | |
| Nurses/Midwives | Ref | | | |
| Doctors/Physicians | 0.55 | 0.11 | [0.33–0.77] | **<0.001**** |
| Pharmacists/Disp Tech | 0.17 | 0.18 | [-0.18–0.53] | 0.350 |
| Allied Health Professionals | 0.34 | 0.10 | [0.14–0.55] | **0.001*** |
| Had Training in IT | | | | |
| No | Ref | | | |
| Yes | 0.66 | 0.20 | [0.26–1.05] | **0.001*** |

*p-value<0.05 and

**p-value<0.001 are statistically significant.

SE—Standard Error; CI—Confidence Interval.

those with less than 5 years. However, healthcare providers with 20 years and above (coefficient = -0.88; *p*-value = 0.012) had a decreasing effect on MoITU in healthcare delivery compared with those with less than 5 years. Concerning professional background, healthcare providers who are doctors/physicians (coefficient = 0.55; *p*-value<0.001) and allied health professionals (i.e. health information officers, biomedical scientists, nutritionist, etc.) (coefficient = 0.34; *p*-value = 0.001) had an increasing influence on MoITU in healthcare delivery compared with those with professionals who are nurses/midwives. Regarding training in IT, the healthcare providers with IT training (coefficient = 0.66; *p*-value = 0.001) had a positive relationship with MoITU in healthcare delivery compared with those without IT training [Table 5].

H7: HCP working experience had a significant influence on MoITU.

H8: HCP profession had a significant influence on MoITU.

H9: HCP education had a significant influence on MoITU.

H12: HCP IT training had a significant influence on MoITU

## Discussions

This study assesses HCPs' perceptions towards the utilization of health information applications in healthcare delivery in the Cape Coast Metropolis of Ghana. The study tested the hypotheses of a possible association between background characteristics (such as IT training, working experience, profession, and education) of HCP and their perception towards the benefits of information technology, the barriers to information technology use, and the motive of information technology use in healthcare delivery. The findings of this study to a large extent will contribute to the generation of further knowledge; and the creation of awareness of the benefits and challenges among different health stakeholders which are expected to help in establishing more realistic interventions to promote the utilization of health information applications in healthcare delivery.

### The benefits of information technology in healthcare delivery

The advantages of health information technology (IT) include facilitating communication between HCPs; improving medication safety, and promoting quality of care through optimized access to and adherence to guidelines. Health IT systems permit the collection of data for use in quality management, outcome reporting, and public health disease surveillance and reporting [19]. Our study found that on average HCPs moderately agreed that there are benefits of information technology use in healthcare delivery. The benefit includes easier to access patient records, easier ways to find investigation results and ensuring patients'privacy. Again, it was found that, on average, IT helps in managing patients, provides speed to accomplish work, saved paperwork, facilitates coordination among departments, improves decisions making process, IT reduces medical errors, improves the quality of patients' care and decreases the workload. This conformed to the other studies of Abdulai and Adane [20,21] which found that IT helps in managing patients, provides speed to accomplish work, saved paperwork and IT reduces medical errors, improves the quality of patients' care and decreases workload. Other studies by Bardhan and Thouin [22] also found that HCPs have a good perception of the benefits of information technology use. Other benefits mentioned by respondents include the facilitation of coordination among departments, improved decisions making process, ensuring patients'privacy, reduced medical errors and improvement of the quality of patients' care. These benefits are not different from what other studies by Cline and Luiz found in developing countries [23].

### Factors influencing Benefits of Information Technology (BoIT) in healthcare delivery

In line with Lluch's study [24], our research reveals a statistically significant correlation between education and the benefits of Information Technology (BoIT) in healthcare delivery. Specifically, healthcare professionals with bachelor's, master's, or other advanced diplomas/degrees demonstrated an increasing positive impact on BoIT compared to those with intermediate certificates. This implies that higher educational attainment enhances their understanding and appreciation of IT advantages in healthcare settings. Additionally, our study supports the findings of Wu et al. [25], indicating that working experience significantly influences BoIT in healthcare delivery. Healthcare providers with 5–9 and 10–14 years of experience exhibited

increasing positive effects on BoIT compared to those with less than 5 years of experience, possibly due to gained exposure to IT benefits over time. Bhattacharya et al.'s systematic review [26] aligns with our findings, indicating that the professional background of healthcare providers significantly influences BoIT. Doctors/physicians and allied health professionals showed a greater influence on BoIT compared to nurses/midwives, implying varied perceptions of IT benefits across healthcare professions. Moreover, our study found a significant positive association between IT training and BoIT in healthcare delivery, consistent with Hossain and Prybutok's research on Electronic health record system implementation [27]. IT training equips healthcare professionals with the necessary knowledge and skills to fully harness IT advantages in their practice.

## The barriers to information technology use

As healthcare moves toward the widespread use of technology to meet growing concerns over patient care and safety, the need to create a health information technology (HIT) infrastructure to transport data and create information-sharing networks between HCPs becomes increasingly apparent [28]. With the repeated calls for the use of technology in healthcare, there are, nevertheless, barriers to its adoption. Our study indicated that respondents slightly agreed that there had been an insufficient number of computers, frequent system breakdowns, low system performance and lack of training for the hospital staff etc. as barriers to the utilization of health information applications. This affirmed other studies by De Leeuw et al. [29] and Abdullah et al. [30] which found that lack of suitable training, lack of technical support and increase in workload were the barriers to the use of information technology. Another study by Ahmad in developing countries found that the lack of availability of computers is a challenge for the adoption of information technology in healthcare institutions [31]. However, findings of a study by Kelchner revealed that the physical location of computers or shortcomings in technical management support and technophobia are major challenges in the use of IT [32].

## Factors influencing healthcare providers' perceived barriers to information technology use

Once again, our study confirms the significant relationship between age and Barriers to Information Technology Use (BITU) in healthcare delivery, as established in Lluch's research [24]. Healthcare providers aged 50 years and above demonstrated a decreasing influence on BITU compared to those aged 20–29 years (coefficient = -0.84; p-value<0.001). This suggests that as healthcare providers age, they may perceive fewer obstacles to IT adoption in healthcare delivery. The older generation of healthcare professionals may have developed strategies (antidotes) over their years of experience to overcome IT-related challenges, enhancing their acceptance of IT integration. In line with Holden and Karsh's findings [33], our study also supports a significant relationship between education and BITU in healthcare delivery. For example, healthcare providers with a master's degree exhibited an increasing influence on BITU compared to those with an intermediate certificate (coefficient = 0.33; p-value = 0.004). Higher levels of education may foster a deeper understanding of the benefits and potential of IT in healthcare, making professionals with advanced degrees more receptive to IT adoption. Moreover, Wu et al.'s study on factors affecting hospital employees' intentions to use mobile learning [25] reinforces our findings, demonstrating a statistically significant relationship between working experience and BITU in healthcare delivery. Healthcare providers with 5–9 years of experience showed a positive association with BITU compared to those with less than 5 years of experience (coefficient = 0.21; p-value = 0.030). Increased working experience may lead to the

successful navigation of barriers to IT use, fostering a more favourable perception of IT integration into their practices.

## The motive to information technology use in healthcare delivery

One of the main obstacles to the successful deployment and adoption of IT in hospitals is the staff's knowledge and attitude about using IT in the facilities, as well as their abilities and the current state of computerization in healthcare facilities [34]. Because of this, HCPs must receive on-the-job training if they are to adopt positive attitudes towards IT use and increase staff confidence [35]. Our study demonstrated that respondents appeared to have a positive motivation for using health information applications in healthcare delivery. Furthermore, this study revealed that providing new/durable applications, staff training, change in hospitals' work procedures, and technical support are the main reasons why information technology is used in healthcare delivery. These findings corroborated with that of Kipturgo et al. [36], which discovered that HCPs have a favourable opinion. Another study conducted in China by Chen et al. revealed that HCPs had a good reason for using information technology in their work [37]. The survey also noted the usage of IT applications that are long-lasting, training of employees, and offering technical assistance. Information technology is used to support health personnel technically, according to studies by Atinga, Abor, Suleman, Anaba, and Kipo in Ghana [38]. This study suggested that there was a significant association between HCPs background characteristics (such as IT training, working experience, profession, and education) and their perception towards the benefits of information technology as well as their motive for information technology use in healthcare delivery.

## Factors influencing Motives of Information Technology Use (MoITU) in healthcare delivery

Cresswell & Sheikh [39], in their interpretative review on organizational issues in health information technology adoption, found a statistically significant relationship between age and MoITU in healthcare delivery, which aligns with our study's findings. Healthcare providers aged 50 years and above exhibited a decreasing influence on MoITU compared to those aged 20–29 years (coefficient = -0.66; p-value = 0.003), suggesting that as healthcare providers age, their motives for using information technology in healthcare may diminish. Older professionals may have developed effective strategies (antidotes) to overcome IT-related challenges, resulting in a perceived lower impact on their motivation for IT utilization. Similarly, Venkatesh et al. demonstrated in their study on User acceptance of information technology [40] that education significantly influences MoITU in healthcare delivery. Our findings align with theirs, as we observed that healthcare professionals with bachelor's, master's, and other professional diploma/degree had an increasing effect on MoITU compared to those with intermediate certificates. Higher education levels may foster a better understanding of the benefits and potential of information technology in healthcare, leading to greater appreciation for its advantages among professionals with advanced degrees. Moreover, working experience significantly influenced MoITU in healthcare delivery. Healthcare providers with 5–9 years of experience had an increasing effect on MoITU compared to those with less than 5 years (coefficient = 0.35; p-value<0.001). However, those with 20 years and above showed a decreasing effect on MoITU compared to those with less than 5 years (coefficient = -0.88; p-value = 0.012), indicating that moderate working experience (5–9 years) positively impacts IT use motives, while very experienced providers may have decreased motives due to prolonged exposure to IT challenges. Furthermore, our study found a significant positive association between IT training and MoITU in healthcare delivery. Healthcare providers with IT training

exhibited a higher perception of the benefits of IT utilization compared to those without IT training (coefficient = 0.66; p-value = 0.001), underscoring the importance of providing training to enhance healthcare professionals' motivation and confidence in utilizing IT in healthcare [27].

### Strengths and limitations of the study

However, the study has some limitations. It employs a cross-sectional design, capturing data at a specific point in time, which limits the ability to establish causal relationships or track changes over time. The small sample size and focus on a single geographical area may restrict the generalizability of the findings to broader populations or other regions. Self-report bias could be present as participants may provide socially desirable responses, potentially impacting the accuracy of the findings. The lack of qualitative data may limit a more in-depth exploration of the contextual nuances influencing healthcare professionals' attitudes toward IT utilization. Furthermore, there may be unmeasured variables and potential confounding factors that were not considered, affecting the study's results. Despite these limitations, the study offers valuable insights for improving the adoption and utilization of health information applications, facilitating better patient care and healthcare outcomes in the Cape Coast Metropolis of Ghana.

## Conclusions

The results of the study reveal that healthcare professionals view IT utilization positively, as it leads to improved communication and patient safety. This highlights the importance of IT in healthcare and raises awareness among policymakers and other stakeholders. Education, work experience, and professional background are factors that significantly influence healthcare professionals' attitudes towards IT, and IT training enhances their perception of its benefits. The study also identifies barriers, such as the lack of resources and training, that need to be addressed to improve the situation. Although the study has some limitations, its implications can guide interventions aimed at promoting the adoption of health information applications, ultimately leading to better patient care in the Cape Coast Metropolis of Ghana.

## Supporting information

**S1 Table. Table 6: Descriptive statistics study constructs.**
(PDF)

**S1 File.**
(XLS)

## Acknowledgments

Our special thanks go to the staff who supported in diverse ways towards the conduct of the study.

## Author Contributions

**Conceptualization:** Richard Okyere Boadu, Nathan Kumasenu Mensah.

**Data curation:** Richard Okyere Boadu, Christiana Dziyaba.

**Formal analysis:** Richard Okyere Boadu, Godwin Adzakpah, Nathan Kumasenu Mensah, Kwame Adu Okyere Boadu.

**Funding acquisition:** Richard Okyere Boadu, Christiana Dziyaba.

**Investigation:** Richard Okyere Boadu.

**Methodology:** Richard Okyere Boadu.

**Project administration:** Richard Okyere Boadu.

**Resources:** Richard Okyere Boadu.

**Software:** Richard Okyere Boadu.

**Supervision:** Richard Okyere Boadu.

**Validation:** Richard Okyere Boadu.

**Visualization:** Richard Okyere Boadu.

**Writing – original draft:** Richard Okyere Boadu, Godwin Adzakpah, Nathan Kumasenu Mensah, Kwame Adu Okyere Boadu, Jonathan Kissi, Christiana Dziyaba, Rosemary Bermaa Abrefa.

**Writing – review & editing:** Richard Okyere Boadu, Godwin Adzakpah, Nathan Kumasenu Mensah, Kwame Adu Okyere Boadu, Jonathan Kissi, Rosemary Bermaa Abrefa.

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
