## [Decision Letter · Decision Letter 0]

14 Jul 2023

PONE-D-23-07152Healthcare Provider’s Perception towards Utilisation of Health Information Applications in Healthcare Delivery in Cape Coast Metropolis, GhanaPLOS ONE

Dear Dr. Okyere Boadu,

Thank you for submitting your manuscript to PLOS ONE. After careful consideration, we feel that it has merit but does not fully meet PLOS ONE’s publication criteria as it currently stands. Therefore, we invite you to submit a revised version of the manuscript that addresses the points raised during the review process.

Authors should kindly consider the comments of both reviewers when revising the manuscript. In particular, attention should be paid to the study design so stated. Testing for hypotheses and determining associations between variables does not make the study a descriptive one. Authors should also ensure that all data for the study are presented.

We look forward to receiving your revised manuscript.

Kind regards,

Gifty Dufie Ampofo, M.D., Ph.D

Academic Editor

PLOS ONE

Journal Requirements:

- https://doi.org/10.1155/2021/5547544

- http://erl.ucc.edu.gh:8080/xmlui/bitstream/handle/123456789/3843/NUKUNU%2c%202018.pdf?isAllowed=y&sequence=1

- https://www.coursehero.com/file/71560624/CloudeMR-A-Cloud-Based-Electronic-Medicapdf/

- https://www.science.gov/topicpages/p/patient+information+sheet.html

- https://digitalcommons.providence.edu/cgi/viewcontent.cgi?referer=&httpsredir=1&article=1031&context=auchs

In your revision ensure you cite all your sources (including your own works), and quote or rephrase any duplicated text outside the methods section. Further consideration is dependent on these concerns being addressed.

Reviewers' comments:

Reviewer's Responses to Questions

**Comments to the Author**

1. Is the manuscript technically sound, and do the data support the conclusions?

Reviewer #1: Yes

Reviewer #2: Partly

2. Has the statistical analysis been performed appropriately and rigorously? 

Reviewer #1: Yes

Reviewer #2: Yes

3. Have the authors made all data underlying the findings in their manuscript fully available?

Reviewer #1: No

Reviewer #2: Yes

4. Is the manuscript presented in an intelligible fashion and written in standard English?

Reviewer #1: Yes

Reviewer #2: Yes

5. Review Comments to the Author

Reviewer #1: COMMENTS TO AUTHORS

Let me congratulate authors for taking time to research into such an important phenomenon. I find the following comments useful for enriching the content of the manuscript.

Abstract

The first appearance of the phrase healthcare provider must be in full, and immediately afterwards the acronym HCP follows in a bracket. It is only when this has been done that subsequent sentences can simply use the acronym HCP. I see an acronym HP under methods – this has not been defined in earlier sentences.

“We conclude that HPs appear to have a positive perception…” – What precisely do you mean by a positive perception?

Introduction

“Many IT initiatives have sprung up including the 78 DHIMS and MoTeCH and NHIA initiatives..”

Be precise about which NHIA initiatives are IT initiatives?

” Waema [11] and Knuth et al. [12], all bid to improve…”

Check this sentence again and correct the construction error.

Research Methods

I. The design should rather be stated as Descriptive Cross-sectional and not cross-sectional descriptive.

II. You obtained an 88% response rate; what are the implications of this on your statistical power? Why was a power analysis not performed and your statistical power reported?

III. Check sentence construction errors under heading “selection of facilities”

IV. You have indicated the use of a simple random sampling technique – my question is, how were you able to interview staff who were selected through your mobile app generator but were either on annual leave, indisposed, etc.?

V. You mention the use of a Cronbach Alpha test to check for internal reliability among items – My question is, can you state examples of some of the precise variables which were being measured by the items? And also state the precise alpha scores obtained from the Cronbach test? And the interpretation of those scores?

VI. Can authors explain why they avoided a qualitative study for a phenomenon like this, and what the possible implications of this will be.

Discussion

Can you also discuss the potential limitations of your study; of course, this cannot be a perfect study!

Reviewer #2: "The study seeks to provide relevant information for policy decisions on the use of health information technology to improve patients’ outcomes. However, there are some issues with it that must be addressed by authors:

1. Title: The title could be revised as follows; Healthcare providers’ perception towards utilization of health information applications in healthcare delivery in health facilities in Cape Coast Metropolis, Ghana

2. Abstract: this must be improved as there are some grammatical errors that must be corrected. The background to the study should be brief- two or three sentences with the problem well projected will help a lot. Under the study conclusions, the implications of the findings should stand out clearly.

3. Introduction: An explanation of ICT and Health Information Technology and it/their forms should be projected. Both global level related evidence and evidence from developing countries including Ghana should be extensively used. Authors should also include the rationale for introducing ICT in the health sector. Previously what were the challenges with the use of paper-based records in the health facilities in the Cape Coast Metropolis? Has the introduction of ICT solved those problems? A description of related studies and their gaps will be required to give a solid foundation for the study.

4. Research methodology: This was not rigorously explained. The strobe approach/checklist should be used to present the findings. Authors should provide detailed information on why the study is a descriptive cross-sectional one. A description of the study site will be helpful. A subheading should be created as a study site and the rationale for doing the study in the three health facilities well described. Also, lines 110 to 120 should be put together and labelled sampling procedures.

5. Results: Both descriptive and inferential results should be presented. Demographic information of the participants should be accompanied by a table. Data analysis should be properly described. Inferential results should be presented.

6. Discussion should be strengthened with related works and implications of the findings extensively described. The study limitations should also be described.

7. References: a lot of work has been done around the use of ICT and clients’ satisfaction in northern Ghana. Authors should look for such studies to strengthen the quality of the work."

Further, please see attached file.

6. PLOS authors have the option to publish the peer review history of their article (what does this mean?). If published, this will include your full peer review and any attached files.

Reviewer #1: No

Reviewer #2: No

---

## [Author Response · Author response to Decision Letter 0]

3 Aug 2023

31 July 2023

Editor-in-Chief

PLOS ONE

Dear Sir/Madam,

Re: Responses to Reviewers and Editor’s Comments

On behalf of my colleagues, I am submitting responses to Reviewers and Editor’s comments raised in our article “Healthcare providers’ perception towards utilization of health information applications and its associated factors in healthcare delivery in health facilities in Cape Coast Metropolis, Ghana”. 

The following areas of concern raised by the reviewers and Editor with responses (highlighted) as detailed below:

Response to the 1st Reviewer

COMMENTS TO AUTHORS

Let me congratulate authors for taking time to research into such an important phenomenon. I find the following comments useful for enriching the content of the manuscript.

Thank you for your feedback.

Abstract

The first appearance of the phrase healthcare provider must be in full, and immediately afterwards the acronym HCP follows in a bracket. It is only when this has been done that subsequent sentences can simply use the acronym HCP. I see an acronym HP under methods – this has not been defined in earlier sentences. 

Thank you for your feedback. We have provided the full meaning of HCP at the first instance (see line 28)

“We conclude that HPs appear to have a positive perception…” – What precisely do you mean by a positive perception?

Introduction

“Many IT initiatives have sprung up including the 78 DHIMS and MoTeCH and NHIA initiatives..”

Be precise about which NHIA initiatives are IT initiatives?

Thank you for your feedback. We revised the manuscript accordingly [see lines 79-92]

” Waema [11] and Knuth et al. [12], all bid to improve…”

Check this sentence again and correct the construction error.

Thank you for your feedback. We revised the manuscript accordingly [see lines 96-98]

Research Methods

I. The design should rather be stated as Descriptive Cross-sectional and not cross-sectional descriptive. 

Thank you for your feedback. We revised the manuscript accordingly [see lines 19 and 124]

II. You obtained an 88% response rate; what are the implications of this on your statistical power? 

Thank you for your feedback. We revised the manuscript accordingly to read to include the 80% power [see line 159].

The implication of the response rate on the power has been included in the manuscript [see lines 211-212]

Why was a power analysis not performed and your statistical power reported?

Thank you for your feedback. We have corrected the omission [see line 159]

III. Check sentence construction errors under heading “selection of facilities”

Thank you for your feedback. We have corrected the error [see line 165]

IV. You have indicated the use of a simple random sampling technique – my question is, how were you able to interview staff who were selected through your mobile app generator but were either on annual leave, indisposed, etc.?

Thank you for your feedback. Staff who were available and selected; and voluntarily consented to participate were included in the study [see line 170]

V. You mention the use of a Cronbach Alpha test to check for internal reliability among items – My question is, can you state examples of some of the precise variables which were being measured by the items? And also state the precise alpha scores obtained from the Cronbach test? And the interpretation of those scores?]

Thank you for your feedback. The manuscript has been revised to include some of the precise variables which were being measured by the items and how they were being measured (table 1). [See lines 187-197]

VI. Can authors explain why they avoided a qualitative study for a phenomenon like this, and what the possible implications of this will be.

Thank you for your feedback. We were constraint with timelines to submit the project work and so could not include the qualitative aspect.

Discussion

Can you also discuss the potential limitations of your study; of course, this cannot be a perfect study!

Thank you for your feedback. We have included limitation to the study [see line 453-462]

Response to the 2nd Reviewer

The study seeks to provide relevant information for policy decisions on the use of health information technology to improve patients’ outcomes. However, there are some issues with it that must be addressed by authors:

1. Title: The title could be revise as follows; Healthcare providers’ perception towards utilization of health information applications in healthcare delivery in health facilities in Cape Coast Metropolis, Ghana

Thank you for your feedback. We have revised the title accordingly [see lines 1-2]

2. Abstract: this must be improved as there are some grammatical errors that must be corrected. The background to the study should be brief- two or three sentences with the problem well projected will help a lot. Under the study conclusions, the implications of the findings should stand out clearly.

Thank you for your feedback. We have included the implication of findings [see lines 41, 469-472]

3. Introduction: Explanation of ICT and Health Information Technology and it/their forms should be projected. Both global level related evidence and evidence from developing countries including Ghana should be extensively used. Authors should also include the rationale for introducing ICT in the health sector. Previously what were the challenges with the use of the paper-based records in the health facilities in the Cape Coast Metropolis? Has the introduction of ICT solved those problems? A description of related studies and their gaps will be required to give a solid foundation for the study.

Thank you for your feedback. We have revised the manuscript to the challenges of the paper-based patients records management and the piloting of electronic health record etc. [see lines 93-105]

4. Research methodology: This was not rigorous explained. The strobe approach/checklist should be used to present the findings. Authors should provide detailed information on why the study is a descriptive cross sectional one. A description of the study site will be helpful. A sub heading should be created as study site and the rationale for doing the study in the three health facilities well described. Also, lines 110 to 120 should be put together and labelled sampling procedures.

Thank you for your feedback. We have included the study sites [see 129-149]. These facilities were selected and included in the study because they have deployed patients’ electronic health record systems to render healthcare delivery to their clients [see lines 164-166]

5. Results: Both descriptive and inferential results should be presented. Demographic information of the participants should be accompanied by a table. Data analysis should be properly described. Inferential results should be presented.

Thank you for your feedback. We have included both descriptive and inferential results [see lines [226-337]

6. Discussion should be strengthened with related works and implications of the findings extensively described. The study limitations should also be described.

Thank you for your feedback. We have revised the manuscript accordingly

7. References: a lot of works have been done around the use of ICT and clients’ satisfaction in northern Ghana. Authors should look for such studies to strengthen the quality of the work.

Best wishes

Thank you for your feedback. We have including additional references to strengthen our work

Thank you for your reconsideration and re-evaluation prior to consideration for publication. My colleagues and I appreciate your time and effort and look forward to hearing from you.

Sincerely,

Richard Okyere Boadu (PhD)

Department of Health Information Management

School of Allied Health Sciences

College of Health and Allied Health Sciences

University of Cape Coast

Cape Coast, Ghana

richard.boadu@ucc.edu.gh

---

## [Decision Letter · Decision Letter 1]

3 Nov 2023

PONE-D-23-07152R1Healthcare providers’ perception towards utilization of health information applications and its associated factors in healthcare delivery in health facilities in Cape Coast Metropolis, GhanaPLOS ONE

Dear Dr. Okyere Boadu,

Thank you for submitting your manuscript to PLOS ONE. After careful consideration, we feel that it has merit but does not fully meet PLOS ONE’s publication criteria as it currently stands. Therefore, we invite you to submit a revised version of the manuscript that addresses the points raised during the review process.

We look forward to receiving your revised manuscript.

Kind regards,

Gifty Dufie Ampofo, M.D., Ph.D

Academic Editor

PLOS ONE

Journal Requirements:

Additional Editor Comments:

Kindly address the comments of the second reviewer for another review.

Reviewers' comments:

Reviewer's Responses to Questions

**Comments to the Author**

1. If the authors have adequately addressed your comments raised in a previous round of review and you feel that this manuscript is now acceptable for publication, you may indicate that here to bypass the “Comments to the Author” section, enter your conflict of interest statement in the “Confidential to Editor” section, and submit your "Accept" recommendation.

Reviewer #1: All comments have been addressed

Reviewer #2: (No Response)

2. Is the manuscript technically sound, and do the data support the conclusions?

Reviewer #1: Yes

Reviewer #2: Yes

3. Has the statistical analysis been performed appropriately and rigorously? 

Reviewer #1: Yes

Reviewer #2: Yes

4. Have the authors made all data underlying the findings in their manuscript fully available?

Reviewer #1: Yes

Reviewer #2: Yes

5. Is the manuscript presented in an intelligible fashion and written in standard English?

Reviewer #1: Yes

Reviewer #2: Yes

6. Review Comments to the Author

Reviewer #1: (No Response)

Reviewer #2: Please, find attached my feedback.

7. PLOS authors have the option to publish the peer review history of their article (what does this mean?). If published, this will include your full peer review and any attached files.

Reviewer #1: No

Reviewer #2: No

---

## [Author Response · Author response to Decision Letter 1]

7 Nov 2023

31 July 2023 

Editor-in-Chief 

PLOS ONE 

Dear Sir/Madam, 

Re: Responses to Reviewers Comments

On behalf of my colleagues, I am submitting responses to the Reviewers’ comments raised in our article “Healthcare providers’ perception towards utilization of health information applications and its associated factors in healthcare delivery in health facilities in Cape Coast Metropolis, Ghana”. 

The following areas of concern were raised by the reviewers and Editor with responses (highlighted) as detailed below:

INTRODUCTION

It is important to describe briefly challenges associated with using ICT for healthcare delivery.

Thank you for the feedback. We have expanded the introduction section to include some challenges associated with using ICT for healthcare delivery [see lines 73-77].

METHODS

Data collection tool

Describe how interviews were done. What languages were used during the interview process, where were interviews done and how many interviews were conducted? The authors can rename the heading as DATA COLLECTION TOOLS AND PROCEDURES.

Thank you for the feedback. We have revised the manuscript accordingly [see lines 177, 185-188].

RESULTS

Line 210, please start with a capital letter; Table 2. Results should be reported in the past. For example, the study revealed that… 

Thank you for the feedback. We have revised the manuscript accordingly [see lines 218].

Discussion

Some typographical errors require redress. Line 360, the word DELIVER should be ‘DELIVERY’.

Thank you for the feedback. We have revised the manuscript accordingly [see line 368].

Thank you for your reconsideration and re-evaluation prior to consideration for publication. My colleagues and I appreciate your time and effort and look forward to hearing from you.

Sincerely,

Richard Okyere Boadu (PhD)

Department of Health Information Management

School of Allied Health Sciences

College of Health and Allied Health Sciences

University of Cape Coast

Cape Coast, Ghana

richard.boadu@ucc.edu.gh

---

## [Editor Report · Decision Letter 2]

4 Jan 2024

Healthcare providers’ perception towards utilization of health information applications and its associated factors in healthcare delivery in health facilities in Cape Coast Metropolis, Ghana

PONE-D-23-07152R2

Dear Dr. Okyere Boadu,

We’re pleased to inform you that your manuscript has been judged scientifically suitable for publication and will be formally accepted for publication once it meets all outstanding technical requirements.

Kind regards,

Gifty Dufie Ampofo, M.D., Ph.D

Academic Editor

PLOS ONE
---

## [Editor Report · Acceptance letter]

24 Jan 2024

PONE-D-23-07152R2 

PLOS ONE

Dear Dr. Okyere Boadu, 

I'm pleased to inform you that your manuscript has been deemed suitable for publication in PLOS ONE. Congratulations! Your manuscript is now being handed over to our production team.

Kind regards, 

on behalf of

Dr. Gifty Dufie Ampofo 

Academic Editor

PLOS ONE